

# Projecting ozone hole recovery using an ensemble of chemistry-climate models weighted by model performance and independence

Matt Amos[1], Paul J. Young[1,2], J. Scott Hosking[3], Jean-François Lamarque[4], N. Luke Abraham[5,6],

Hideharu Akiyoshi[7], Alexander T. Archibald[5,6], Slimane Bekki[8], Makoto Deushi[9], Patrick Jöckel[10],

Douglas Kinnison[4], Ole Kirner[11], Markus Kunze[12], Marion Marchand[8], David A. Plummer[13],

David Saint-Martin[14], Kengo Sudo[15, 16], Simone Tilmes[4], and Yousuke Yamashita[16]

[1]Lancaster University, Lancaster, UK

[2]Centre for Excellence in Environmental Data Science, Lancaster University, Lancaster, UK

[3]British Antarctic Survey, Cambridge, UK

[4]National Center for Atmospheric Research (NCAR), Boulder, Colorado, USA

[5]Department of Chemistry, University of Cambridge, Cambridge, UK

[6]National Centre for Atmospheric Science (NCAS), UK

[7]National Institute of Environmental Studies (NIES), Tsukuba, Japan

[8]LATMOS, Institut Pierre Simon Laplace (IPSL), Paris, France

[9]Meteorological Research Institute (MRI), Tsukuba, Japan

[10]Institut für Physik der Atmosphäre, Deutsches Zentrum für Luft- und Raumfahrt (DLR), Oberpfaffenhofen, Germany

[11]Steinbuch Centre for Computing, Karlsruhe Institute of Technology, Karlsruhe, Germany

[12]Institut für Meteorologie, Freie Universität Berlin, Berlin, Germany

[13]Environment and Climate Change Canada, Montréal, Canada

[14]CNRM, Université de Toulouse, Météo-France, CNRS, Toulouse, France

[15]Graduate School of Environmental Studies, Nagoya University, Nagoya, Japan

[16]Japan Agency for Marine-Earth Science and Technology (JAMSTEC), Yokohama, Japan

**Correspondence:** Matt Amos (m.amos1@lancaster.ac.uk)





**Abstract.** The current method for averaging model ensembles, which is to calculate a multi model mean, assumes model independence and equal model skill. Sharing of model components amongst families of models and research centres, conflated by growing ensemble size, means model independence cannot be assumed and is hard to quantify. We present a methodology to produce a weighted model ensemble projection, accounting for model performance and model independence. Model weights are calculated by comparing model hindcasts to a selection of metrics chosen for their physical relevance to the process or phenomena of interest. This weighting methodology is applied to the Chemistry-Climate Model Initiative (CCMI) ensemble, to investigate Antarctic ozone depletion and subsequent recovery. The weighted mean projects an ozone recovery to 1980 levels, by 2056 with a 95 % confidence interval (2052–2060), 4 years earlier than the most recent study. Perfect model testing and out-of-sample testing validate the results and show a greater projective skill than a standard multi model mean. Interestingly, the construction of a weighted mean also provides insight into model performance and dependence between the models. This weighting methodology is robust to both model and metric choices and therefore has potential applications throughout the climate and chemistry-climate modelling communities.

## 1   Introduction

Global chemistry-climate models (CCMs) are the most comprehensive tools to investigate how the global composition of the atmosphere develops, both naturally and under anthropogenic influence (Flato et al., 2014; Morgenstern et al., 2017; Young et al., 2018). As with projecting climate change, consensus views of the past and potential future evolution of atmospheric composition are obtained from coordinated CCM experiments (Eyring et al., 2008; Lamarque et al., 2013; Morgenstern et al., 2017) and subsequent analysis of the ensemble of simulations (Iglesias-Suarez et al., 2016; Dhomse et al., 2018). Although not a complete sample of structural and epistemic uncertainty, these ensembles are an important part of exploring and quantifying drivers of past and future change, and evaluating the success of policy interventions, such as stratospheric ozone recovery resulting from the Montreal Protocol and its amendments (Dhomse et al., 2018; WMO, 2018). Typically, analysis of an ensemble investigates the behaviour and characteristics of the multi-model mean and the inter-model variance (Solomon et al., 2007; Tebaldi and Knutti, 2007; Butchart et al., 2010), rather than accounting for individual model performance or lack of model independence (Knutti, 2010; Räisänen et al., 2010). Methods to address these shortcomings have been proposed for simulations of the physical climate (Gillett, 2015; Knutti et al., 2017; Abramowitz et al., 2019, e.g.), but this topic has received less attention in the atmospheric composition community. Here, we demonstrate a weighting method for the CCM simulation of Antarctic ozone loss and projected recovery, where the weighting accounts for model skill and independence over specified metrics relevant to polar stratospheric ozone. We apply this to the recent Chemistry-Climate model initiative (CCMI) (Morgenstern et al., 2017) ensemble and demonstrate the impact of the weighting on estimated ozone hole recovery dates.

Many years of scientific studies and assessments have tied stratospheric ozone depletion to the anthropogenic emission and subsequent photochemistry of halogen-containing gases, such as chlorofluorcarbons (CFCs), hydrofluorocarbons (HCFCs) and halons (WMO, 2018). This science guided the development of the Montreal Protocol, and its subsequent amendments, to limit and ban the production of these ozone-destroying gases, and stratospheric ozone is now thought to be recovering (Solomon





et al., 2016; Chipperfield et al., 2017; Ball et al., 2018). Of particular concern is the Antarctic "ozone hole": a steep decline in

high latitude stratospheric ozone during austral spring that can reduce ozone concentrations to near zero at particular altitudes, driven by polar night-time chemistry, cold temperatures and heterogeneous catalysis on polar stratospheric clouds (PSCs) (Solomon, 1999). While the ozone hole continues to appear in each austral spring, it appears be showing signs of recovery (Langematz et al., 2018). The strong cooling associated with Antarctic ozone depletion (Thompson and Solomon, 2002; Young et al., 2012) has driven circulation changes in the stratosphere and in the troposphere, particularly in austral summer. This has

notably included an acceleration and poleward movement of the southern high latitude westerly winds and associated storm tracks (Son et al., 2008; Perlwitz et al., 2008), leading to summertime surface climate changes through many lower latitude regions including the tropics (Thompson et al., 2011).

The recovery process is slow due to the long atmospheric lifetimes of ozone depleting substances, and could be hampered by releases of ozone depleting substances (ODSs) not controlled by the Montreal Protocol, such as short-lived halogens (Claxton

et al., 2019; Hossaini et al., 2019) or nitrous oxide (Portmann et al., 2012; Butler et al., 2016), or instances of non-compliance, such as the recent fugitive emissions of CFC-11 (Montzka et al., 2018; Rigby et al., 2019). Recovery itself is often defined as the date at which the ozone layer returns to its 1980 levels, and this is the benchmark used by the WMO (WMO, 2018) to assess the progress due to the implementation of the Montreal Protocol.

The assessment of when the ozone layer will recover is conducted using an ensemble of chemistry-climate models, forced

by past and projected future emissions of ozone depleting substances (ODSs) and climate (Eyring et al., 2010; Dhomse et al., 2018). Such ensembles are used to establish the robustness of the model results for a particular scenario: when several models agree, the prevailing assumption is that we can have greater confidence in the model projections. Yet, there has been much discussion about how true this assumption is (Tebaldi and Knutti, 2007; Sanderson et al., 2015b; Abramowitz et al., 2019). In an ideal scenario, every model within an ensemble would be independent and have some random error. In this case, we

would expect that increasing the ensemble size would decrease the ensemble uncertainty and allow us to better constrain the mean value. However, in modern model inter-comparison projects this is not the case: although often developed independently, models are not truly independent, often sharing components and parametrisations (Knutti et al., 2013); models are not equally good at simulating the atmosphere (Reichler and Kim, 2008; Bellenger et al., 2014); and lastly, models do not have a predictable random error but instead have layers of uncertainty extending from uncertainties in parametrising sub-grid processes (Rybka

and Tost, 2014) to structural uncertainties from the design of the model (Tebaldi and Knutti, 2007; Knutti, 2010).

Given these issues, there is currently no consensus on how best to combine model output when analysing an ensemble. Probably the most widely used and simplest is to take a multi model mean where each model contributes equally, and indeed it has also been established that an ensemble mean performs better than any single model (Gleckler et al., 2008; Reichler and Kim, 2008; Pincus et al., 2008; Knutti et al., 2010). A more sophisticated method is to weight individual ensemble members,

accounting for model performance as well as the degree of a model's independence. Weighting methods of various forms have been developed and implemented on global physical climate model ensembles (Tebaldi et al., 2005; Räisänen et al., 2010; Haughton et al., 2015; Knutti et al., 2017), but seldom for atmospheric composition. In most cases the weights are calculated from comparison of model hindcasts to observational data, either for a single variable of interest or over a suite of diagnostics.





In addition to these weighting techniques there are other methods for generating ensemble means, such as clustering (Yuan and Wood, 2012; Hyde et al., 2018) and reliability ensemble averaging (REA) (Giorgi and Mearns, 2002). The main motivation for using a weighted mean is to encapsulate model skill and model independence, such that we down-weight models which perform less well and/or are more similar.

Quantifying model skill (or performance) against comparable observations forms an important part of the validation and analysis of multi-model ensembles (Gleckler et al., 2008; Flato et al., 2014; Harrison et al., 2015; Hourdin et al., 2017; Young et al., 2018). Many CCM inter-comparison projects feature validation and assessment through the use of observation-based performance metrics, which may capture model performance for particular atmospheric variables (e.g., temperature, chemical species concentrations, jet position), or be a more derived quantity which gets closer to evaluating the model against the process it is trying to simulate (e.g., ozone trends vs. temperature trends, chemical species correlations, chemistry-meteorology/transport relationships) (Eyring et al., 2006; Waugh and Eyring, 2008; Christensen et al., 2010; Lee et al., 2015). Performance metrics are chosen based upon expert knowledge of the modelled system to ensure that metrics are highly related to the physical or chemical processes that the models are being evaluated on.

In this study we develop a weighting methodology, originally presented by Sanderson et al. (2017) and Knutti et al. (2017), for CCM ensembles that accounts for model performance and model independence. We apply it to the important issue of estimating Antarctic ozone recovery using several well-established metrics of model performances, where previously only unweighted means have been used. We first describe our weighting framework in Sect. 2, before describing the model and observational data in Sect. 3. Section 4 presents the application of the weighting framework to Antarctic ozone depletion and the corresponding results. Sections 5 and 6 present a summary and our conclusions.

## 2 The model weighting framework

In this study, we develop and exploit a framework to calculate model weights based on the recent work in the physical climate science community (Sanderson et al., 2015a, b, 2017; Knutti et al., 2017). Here, for an ensemble of $N$ models, the weight for model $i$ ($w_i$) is given by

$$w_i = \exp\left(-\frac{D_i^2}{n_i \sigma_D^2}\right) \Big/ \left(1 + \sum_{i \neq j}^{N} \exp\left(-\frac{S_{ij}^2}{n_i \sigma_S^2}\right)\right). \tag{1}$$

The numerator captures the closeness of the model to observations. $D_i^2$ is the squared difference between a model and the corresponding observation, which is a measure of performance. The denominator captures the closeness of a model to all other models by comparing the squared difference between them ($S_{ij}^2$). Both $\sigma_D$ and $\sigma_S$ are constants which allow tuning of the weighting to preference either independence or performance (see discussion below). Put more simply, a model has a larger weighting if it closely matches observations and is suitably different to the other models in the ensemble. Finally, Eq. (1) differs from similar versions (e.g., Knutti et al., 2017) through the addition of $n_i$, which is the size of the data used to create the weighting. This could be the amount of grid points for a spatial field, the number of points in a time series, or just one for





a single-valued statistic, and it normalises the data by length allowing for comparison between models and variables with time series of different length and time invariant parameters.

Investigating and evaluating a phenomenon or complex process often relies on identifying multiple metrics since it can only be partially expressed by any single variable. Expert understanding of the physical process is needed to select a set of relevant metrics with which to develop the process-based weighting. Including multiple metrics, provided they are not highly correlated,

has the further benefit of giving less weight to models which perform well but do so for the wrong reasons. In this framework, ensuring that these metrics influence the weighting proportionally is done by normalising the model data using a min-max scaling between $0$ and $1$.

When combining multiple metrics into a weighting, the weight of the $i^{\text{th}}$ model can be found from

$$w_i = \left( \sum_{k=1}^{M} \exp\left( -\frac{D_{ik}^2}{n_{ik}\sigma_D^2} \right) \right) \Big/ \left( M + \sum_{k=1}^{M} \sum_{j\neq i}^{N} \exp\left( -\frac{S_{ijk}^2}{n_{ik}\sigma_S^2} \right) \right), \qquad (2)$$

where $M$ is the total number of metrics and $k$ is the index of the metric. Note that the summation is performed separately over the numerator and the denominator. This means that we calculate the performance and independence scores over all the metrics combined before merging the scores to create the final weighting which, as before, is normalised over all the models to sum to 1.

We take the combined weights for each model and apply them to our parameter or process of interest (the evolution of

stratospheric ozone here). As with the metrics this parameter needn't be a time series and could be a spatial distribution or a single measure. The weighted projection is therefore $x = \sum_{i=1}^{N} w_i x_i$, where $x_i$ is an individual model projection and $w_i$ is the associated weight.

## 2.1 Choosing sigma values

The two scaling parameters ($\sigma_S$, $\sigma_D$) represent a length scale over which two models, or a model and observation, are deemed

to be in good agreement. For example, a large $\sigma_S$ would spread weight over a greater number of models as more models would lie within the length scale of $\sigma_S$. On the other hand, a small $\sigma_S$ sets a higher tolerance for measuring similarity. The choice of the sigma values needs to be considered carefully to strike a balance between weighting all models equally, thus returning to a multi model mean, versus weighting just a few selected models. As the same values of sigma apply across all metrics it is necessary for the data to be normalised to the same values, ensuring that metrics impact the weightings equally. Figure 1 shows

how the weighting function depends on $\sigma_S$, $\sigma_D$, model performance and model independence.

## 3 Applying the weighting framework to the Antarctic ozone hole

We demonstrate the applicability of this weighting framework by applying it to the important and well-understood phenomenon of the Antarctic stratospheric 'ozone hole', for which we can use several decades of suitable observations to weight the models.



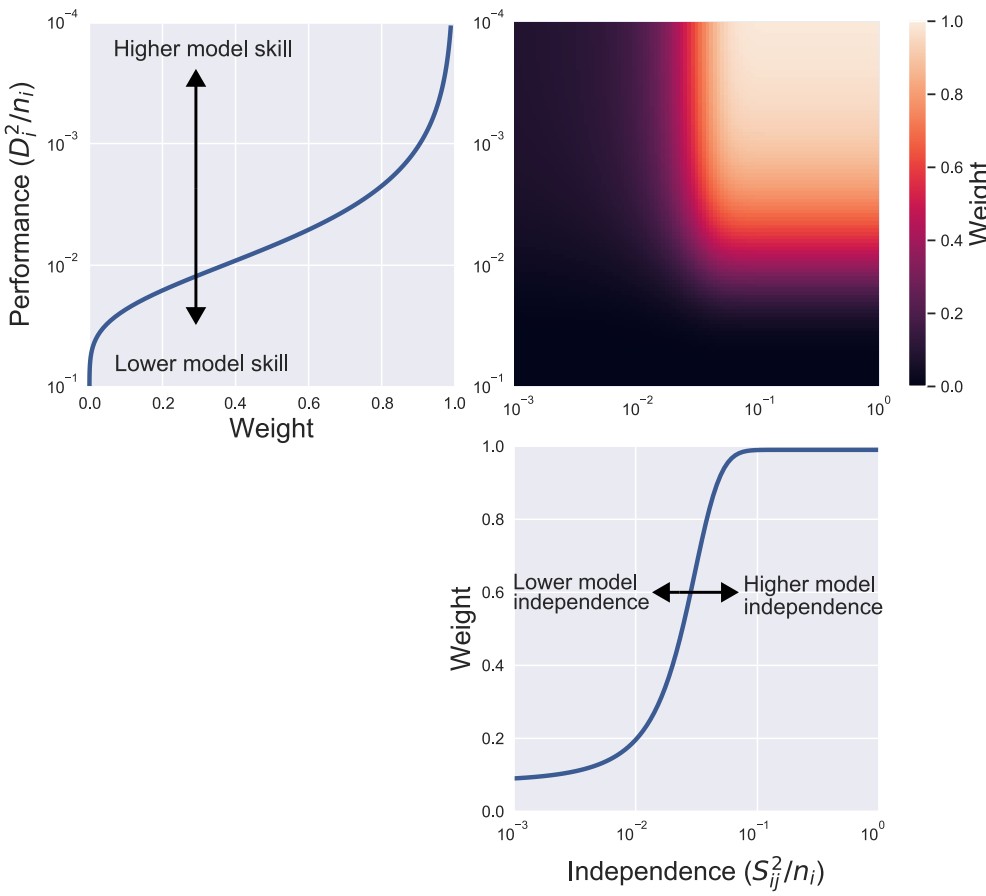

**Figure 1.** Top right shows the overall weighting function $w_i$ (Eq. 1), plotted for 11 models ($N = 11$) with $\sigma_D = 0.1$ and $\sigma_S = 0.1$. Top left shows the contribution to the weighting due to model performance (at $S_{ij}^2/n_i = 1$) and bottom right show the contribution due to model independence (at $D_i^2/n_i = 10^{-4}$). A model which has higher independence and higher skill receives a larger weight. For the weight due to performance (top left) we can see that the weight equals $e^{-1}$ when $D_i^2/n_i = \sigma_D^2$. This shows how $\sigma_D$ acts as a length scale that determines how close a model has to be to observations to receive weight. $\sigma_S$ works similarly, setting the length scale that determines similarity.

Below, we describe the model and observation data used and the metrics selected, against which we measure model perfor-
mance and independence.

## 3.1 Model and observation data sources

CCM output was taken from the simulations conducted under Phase 1 of the Chemistry-Climate Model Initiative (CCMI) ((Morgenstern et al., 2017) and refs. therein), which represents an ensemble of 20 state-of-the-art CCMs (where chemistry and atmospheric dynamics are coupled) and chemistry transport models (CTMs, where the dynamics drives the chemistry, but
there is no coupling). A detailed description of the participating models is provided by Morgenstern et al. (2017), and here we



**Table 1.** The CCMI model simulations used in this analysis and their key references.

| Model | refC1SD realisation(s) | refC2 realisation(s) | Reference(s) |
|---|---|---|---|
| CCSRNIES-MIROC3.2 | r1i1p1 | r1i1p1 | Imai et al. (2013), Akiyoshi et al. (2016) |
| CESM1-CAM4Chem | r1i1p1 | r1i1p1 | Tilmes et al. (2015) |
| CESM1-WACCM | r1i1p1 | r1i1p1 | Marsh et al. (2013), Solomon et al. (2015), Garcia et al. (2017) |
| CHASER-MIROC-ESM | r1i1p1 | r1i1p1 | Sudo et al. (2002), (Sudo and Akimoto, 2007), Watanabe et al. (2011), Sekiya and Sudo (2012) Sekiya and Sudo (2014) |
| CMAM | r1i1p1 | r1i1p1 | Jonsson et al. (2004), Scinocca et al. (2008) |
| CNRM-CM5-3 | r1i1p2 r2i1p2[a] | r1i1p1 | Michou et al. (2011), Voldoire et al. (2013) |
| EMAC-L47MA | r1i1p1 r1i1p2[a] | r1i1p1 | Jöckel et al. (2010), Jöckel et al. (2016) |
| EMAC-L90MA | r1i1p1 r1i1p2[a] | r1i1p1 | |
| IPSL | r1i1p1 | r1i1p1 | Marchand et al. (2012), Szopa et al. (2013), Dufresne et al. (2013) |
| MRI-ESM1r1 | r1i1p1 | r1i1p1 | Deushi and Shibata (2011), Yukimoto (2011), Yukimoto et al. (2012) |
| UMUKCA-UCAM | r1i1p1 | r1i1p1 | Morgenstern et al. (2009), Bednarz et al. (2016) |

[a] Represents the simulations used in the similarity analysis, but that did not form part of the model weighting.

briefly review their overarching features. Most models feature explicit tropospheric chemistry and have a similar complexity of stratospheric chemistry though there is some variation in the range of halogen source gases modelled. Horizontal resolution of the CCMs ranges from between $1.125° \times 1.125°$ to $5.6° \times 5.6°$. Vertically, the atmosphere is simulated from the surface to near the stratopause by all models, and many also resolve higher in the atmosphere. Vertical resolution varies throughout the models, both in the number of levels (34 to 126) and their distribution. All models simulate the stratosphere, although they differ in whether they have been developed with a tropospheric or stratospheric science focus.

We focus on two sets of simulations, called refC1SD and refC2, and for the weighting analysis we only consider models which ran both simulations. Table 1 details the exact model simulations used. The refC1SD simulations cover 1980–2010 and represent the specified dynamics hindcast, where the models' meteorological fields are nudged to reanalysis datasets in order that the composition evolves more in line with the observed inter-annual variability of the atmosphere. In addition to being nudged by meteorology the refC1SD runs are forced by realistically varying boundary conditions, including greenhouse gas (GHG) concentrations, ODS emissions, and sea surface temperatures (SSTs) and sea-ice concentrations (SICs). The refC1SD simulations are used to create the model weightings since these are the models' best attempt at replicating the past, giving reasonable confidence that any down-weighting arises due to poorer model performance or strong inter-model similarity. It must be noted that the nudging process is not consistent across the models (Orbe et al., 2018) and we should be mindful that it has the capability to influence the weighting. We discuss the choice to use refC1SD simulations in greater detail in Sect. 5.





**Table 2.** The observational products and respective variables used to construct metrics on which to weight the models.

| Product | Variable | Metric/s | Citation |
|---------|----------|----------|----------|
| MSU | Lower stratosphere temperature (TLS) | TLS/TLS Gradient/Ozone-temperature | Mears and Wentz (2009) |
| NIWA-BS | Total Column Ozone V3.4 (TCO) | TCO gradient/Ozone-temperature | Bodeker et al. (2005) |
| GOZCARDS | Hydrogen chloride concentration | Antarctic hydrogen chloride concentration | Froidevaux et al. (2015) |
| ERA-Interim | Eastward wind speed | Polar vortex breakdown trend | Berrisford et al. (2011) |

The refC2 simulations cover 1960–2100 and are used to construct weighted projections of Antarctic ozone recovery, using the weights calculated from refC1SD. The forcing from GHGs and anthropogenic emissions follows the historical scenario conditions prescribed for the fifth coupled model Inter-comparison project (CMIP5) (Lamarque et al., 2010) up to the year 2000, and subsequently follows representative concentration pathway (RCP) 6.0 for GHGs and tropospheric pollutant emissions (van Vuuren et al., 2011); the ODS emissions follow the World Meteorological Organisation (WMO) A1 halogen scenario (WMO, 2011). From CCMI this is the only scenario which estimates the future climate change and developments to stratospheric ozone.

Model performance was evaluated against a series of well-accepted metrics (see below), drawing from widely used observational and reanalysis datasets listed in Table 2. Assessing models and ensembles using observational data is a principal way of validating models (Eyring et al., 2006; Waugh and Eyring, 2008; Dhomse et al., 2018) and this is the methodology we follow, with the addition that we create the weights based upon this skill, alongside model independence.

Like many ozone recovery studies, we utilise TSAM (time series additive modelling) (Scinocca et al., 2010) to quantify projection confidence, which produces smooth estimates of the ozone trend whilst extracting information about the inter-annual variability. Here, the TSAM procedure involves finding individual model trends for the refC2 simulations by removing the inter-annual variability using a generalised additive model. Each model trend is then normalised to its own 1980 value. The weighted mean (WM) is created by summing model weights with individual model trends. Two uncertainty intervals are created: a 95 % confidence interval, where there is a 95 % chance that the WM lies within; and a 95 % prediction interval, which captures the uncertainty of the WM and the inter-annual variability.

## 3.2 Metric choices - How best to capture ozone depletion

The first step in the weighting process is to identify the most relevant processes that affect Antarctic ozone depletion to allow for appropriate metric choice. Suitable metrics require adequate observational coverage and for the models to have outputted the corresponding variables. The metrics we chose are as follows:

**Total ozone column gradient**. This is the first derivative with respect to time of the total ozone column. Given the discontinuity in the total ozone column record, the years 1992–1996 are excluded. It is a southern polar cap (60°S–90°S) average over austral spring (October and November). September is not included due to discontinuous coverage in the observations.

**Lower stratosphere temperature**. The lower stratosphere temperature for all of the models are constructed using the MSU TLS-weighting function. The MSU dataset extends to 82.5°S, and therefore the southern polar cap average ranges from 60°S




to 82.5°S and is temporally averaged over austral spring (Sept, Oct, Nov).

**Lower stratosphere temperature gradient**. This is the first derivative with respect to time of the lower stratospheric temperature found above.

**Breakdown of the polar vortex**. The vortex breakdown date is calculated as when the zonal mean wind at 60°S and 20 hPa transitions from eastward to westward as per Waugh and Eyring (2008). We find the trend of the breakdown date between the years 1980–2010 and the gradient of the trend forms the polar vortex breakdown metric.

**Ozone-temperature gradient**. Both the lower stratosphere temperature and the ozone are separately averaged over 60°S to 82.5°S and the October and November mean was taken. We determined a linear relationship between temperature and ozone and the gradient of this linear relationship forms the ozone-temperature metric (Young et al., 2013).

**Ozone trend-temperature trend gradient**. This is similar to the metric above except that we first calculated the time derivative of the ozone and temperature polar time series before calculating the linear relationship. The gradient of the linear relationship
is the ozone trend temperature trend gradient metric.

**Hydrogen chloride**. The hydrogen chloride concentration was averaged over the austral spring months, throughout the stratosphere, and over the Southern Polar cap, for areas which have observational coverage.

These metrics capture two of the main features of ozone depletion, namely: 1) the decrease in temperature over the poles
caused by the depletion of ozone, and 2) the breakdown of the vortex which has a major role of isolating the ozone depleted air mass. The chlorine metric encapsulates the anthropogenic release of ODSs and the main chemical driver of ozone depletion. Ozone-temperature metrics allow us to look at model success in reproducing the temperature dependency in ozone reaction rates and stratospheric structure. By looking at the instantaneous rate of change as well as the overall trends, we can gather a picture of both short-term and long-term changes for a range of chemical and dynamical processes.

The metrics are not high correlated, except of the total ozone column gradient and the lower stratosphere temperature gradient, which are correlated because of the strong coupling of ozone and temperature in the stratosphere (e.g., Thompson and Solomon, 2008). Although this could be cause to discard one of the metrics, to avoid potential double counting, we retain and use both to weight because the models may not necessarily demonstrate this coupling that we see in observations. By considering this variety of metrics, the approach aims to demonstrate that models do not just get the 'right' output, but that
they do so for the right reasons.

### 3.3    Evaluating the weighting framework

Two types of testing were used to investigate the usefulness of the weighted prediction and to validate metric choices. Firstly, we performed a simple out-of-sample test on the weighted prediction against the total ozone column observations from NIWA-BS. Although the weights are generated from comparison between the specified dynamics runs (refC1SD) and observations,
it does not necessarily follow that the weighted projection created using the free running (refC2) runs will be a good fit for the observations. To test this, we compared the refC2 multi model mean and weighted projection to the observations. Due to the large inter-annual variability in the total column ozone (TCO) observations, we do not expect the weighted average to be





a perfect match; after all, free running models are not designed to replicate the past. However, we need to test the level of agreement between the weighted mean and the observations for an out-of-sample period (2010–2016). This serves a secondary purpose of determining transitivity between the two model scenarios used: i.e., that the weightings found from refC1SD apply to refC2.

Secondly, we used a perfect model test (also known as model-as-truth or a pseudo model test) to determine whether our weighting methodology is producing valid and robust projections. In turn, each model is taken as the pseudo truth and weightings are found in the same way as described in Sect. 2 except the pseudo truth is used in place of observations. From these weightings we can examine the skill with which the weighted mean compares to the pseudo truth. We are normally limited to a single suite of observations, but a perfect model test allows us to test our methodology numerous times using different pseudo truths, demonstrating robustness.

Perfect model testing also allows us to test transitivity between scenarios since, unlike with the obvious temporal limit on observations, the pseudo truth exists in both the hindcast and forecast. If a weighting strategy produces weighted means which are closer to the pseudo truth than a multi model mean, then we can have some confidence that we can apply a weighting across model scenarios. Herger et al. (2019) compare the perfect model test to the cross validation employed in statistics, but note that although necessary, perfect model tests are not sufficient to fully show out-of-sample skill which in this case is scenario transitivity. It should be backed up by out-of-sample testing as described above.

## 4 Applying the weighting framework to Antarctic ozone simulations

### 4.1 Antarctic ozone and recovery dates

Figure 2 shows the October weighted mean (WM) total column ozone (TCO) trend from the refC2 simulations for the Antarctic (60–90°S). The weights are calculated using Eq. (2), and are based on both model performance and independence. All models simulate ozone depletion and subsequent recovery but with large discrepancies in the absolute TCO values and the expected recovery to 1980 levels (see Dhomse et al., 2018), from here on referred to as D18. The WM and multi model mean (MMM) are similar, given the small number of models considered from the ensemble ($N = 11$). At maximum ozone depletion, around the year 2000, the WM projects a significantly lower ozone concentration (5 DU) than the MMM. This steeper ozone depletion seen in the WM fits the observations better than the MMM, although the modelled inter-annual variability seems to under predict the observations.

The WM predicts a return to 1980 TCO levels by 2056 with a 95 % confidence interval (2052–2060). For comparison the recovery dates presented in D18 were 2062 with a $1\sigma$ spread of (2051–2082). Although taken from the same model ensemble (CCMI), the subset of models in this analysis is smaller than that used in D18 meaning that difference in recovery dates between the two works is attributable to both the methodology and the models considered. The smaller number of models used in this study could lead to a narrower confidence interval than the one reported in D18.

The confidence interval for recovery dates is formed from the predictive uncertainty in the WM from the TSAM (for which the 95 % confidence interval is 2054–2059) and the uncertainty associated with the weighting process. Choices made about





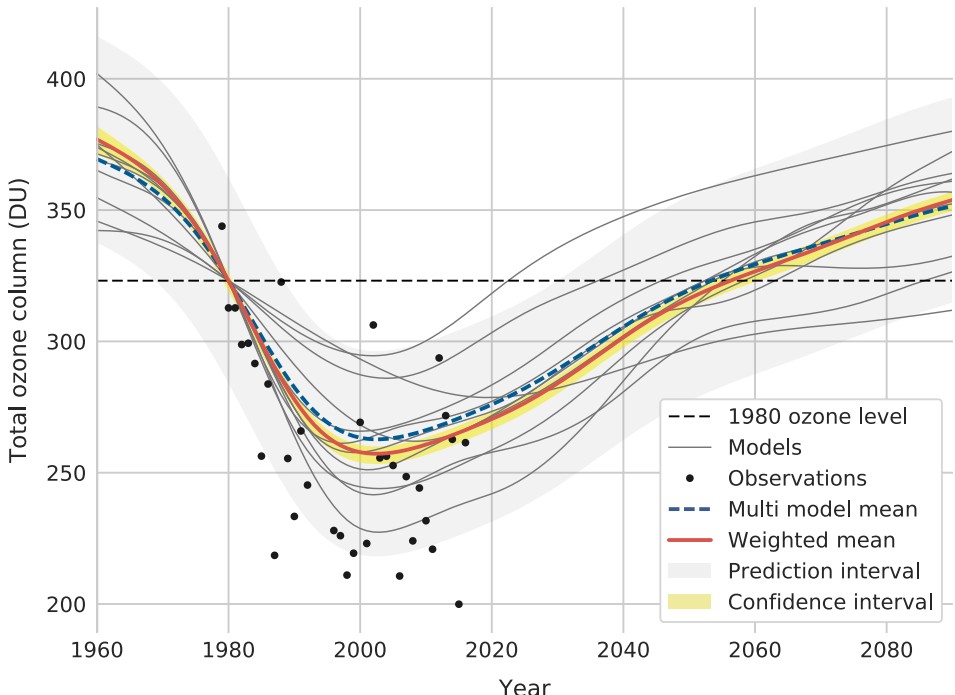

**Figure 2.** Antarctic (60–90°S) October TCO. The weighted mean (refC2 simulations weighted upon refC1SD performance and independence) is shown in red, the multi model mean (refC2 simulations) is shown in blue, and individual refC2 model trends are shown in grey. The NIWA-BS observations are shown in black. All model projections and ensemble projections are normalised to the observational 1979–1981 mean shown as the black dashed line. 95 % confidence and prediction intervals for the weighted mean are also shown with shading.

which models and metrics to include influence the return dates and therefore introduce uncertainty. This is similar to the concept of an "ensemble of opportunity", which is that only modelling centres with the time, resources or interest take part in certain model ensembles. To quantify this uncertainty, we performed a dropout test where a model and a metric were systematically left out of the recovery date calculation. This was done for all combinations of models ($N = 11$) and metrics ($M = 7$), providing

a range of 77 different recovery dates between 2052 and 2058. Combining the TSAM and dropout uncertainties produces a 95 % confidence interval of 2052–2060.

Figure 2 shows the model weights for individual metrics and in total as found using Eqs. (1) and (2). Good agreement is shown between the models for the metrics of lower stratospheric temperature, the temperature gradient, and the TCO gradient. There is one exception of UMUKCA-UCAM which exhibits a colder pole compared to the ensemble and observations.

Resultantly, UMUKCA-UCAM is down-weighted for its lower performance at replicating the historic lower stratospheric temperature. Dissimilarity to the rest of the ensemble will contrastingly increase the weighting but to a lesser effect than the down weighting for performance, due in part to the values of the sigma parameters. In spite of a bias in absolute lower stratospheric temperature, UMUKCA-UCAM does reproduce the trend in the lower stratospheric temperature with similar skill to the other models.





Due to the nudging of temperature that takes place in most of the specified dynamics simulations, we would expect strato-
spheric temperatures to be reasonably well simulated. However, variation exists in nudging methods in addition to inter model
differences and this leads to part of the variability in weights (Orbe et al., 2018; Chrysanthou et al., 2019). For the ozone-
temperature metrics, which although formed from variables linked to nudged fields are more complex in their construction,
we see a much less uniform spread of weights. Furthermore, for processes not directly linked to nudged variables (hydrogen
chloride, ozone, and the polar vortex breakdown trend) there is much less agreement between models. This is captured in the
weights of these metrics which show just a few models possessing large weights.

    The total weighting, formed from the summation of individual metric weights, is largely influenced by CNRM-CM5-3,
which has a weight of 0.27. The CNRM-CM5-3 simulations are more successful at simulating metrics whilst being reasonably
independent from other models, leading to a weight with greater prominence than the other models. This does not mean that
CNRM-CM5-3 is the most skilful model. For example, if two nearly identical models had the highest performance, their final
weights would be much lower as they would be down-weighted for their similarity. All models are contributing towards the
weighted ensemble mean providing confidence that our weighting methodology is not over-tuned and returning model weights
of zero. The lowest total model weight is 55 % the value of a uniform weighting.

## 4.2   Testing the methodology

We performed a perfect model test (Sect. 3.3) to assess the skill of the weighted mean projection, the results of which are
shown in Fig. 4. The perfect model test shows that, on average, using this weighting methodology produces a WM which is
closer to the 'truth' than the MMM by 1 DU. In addition to improvements in projections, the pseudo recovery dates are better
predicted on average, with a maximal improvement of 6 yr.

    Three models, when treated as the pseudo truth, do not show an improvement of the WM with respect to the MMM. Note
that this is not poor performance of the model in question, rather that the weighting methodology does not do an adequate job
of creating a weighted projection for that model as the pseudo truth. Using CHASER-MIROC-ESM as the pseudo truth gives
a worse WM projection than if we used the MMM. However, the average correlation between the CHASER-MIROC-ESM-
simulated TCO and other models in the ensemble is the lowest at 0.65, compared to the average ensemble cross correlation
score of 0.81. Since a weighted mean is a linear combination of models in the ensemble, it is understandable that models with
low correlation to CHASER-MIROC-ESM will be less skilful at replicating its TCO time series. This is why an improvement
is not seen for CHASER-MIROC-ESM as the pseudo truth in the perfect model testing.

    We also performed out-of-sample testing on the WM projection for the years 2010–2016 by comparing it to the TCO
observational time series which was smoothed as described in Sect. 3.1 to remove inter annual variability. The root mean
squared error (RMSE) was used as the metric for goodness of fit. This range of years is chosen as it is the overlap between
the TCO observations, and the years not used in the creation of the weighting. The RMSE of the WM is on average 202 $DU^2$
less than the MMM per year and the RMSE values were 1510 $DU^2$ and 2720 $DU^2$ for the WM and MMM respectively for the
out-of-sample period.





**Figure 3.** Model weights for each of the seven metrics are all shown in blue. The weights account for both performance and independence and are found using Eq. (1). The total weights, as found from Eq. (2), are shown in red and were the weights used to construct the weighted mean shown in Fig. 2. The black dashed line indicates a uniform weighting as prescribed by a multi model mean.





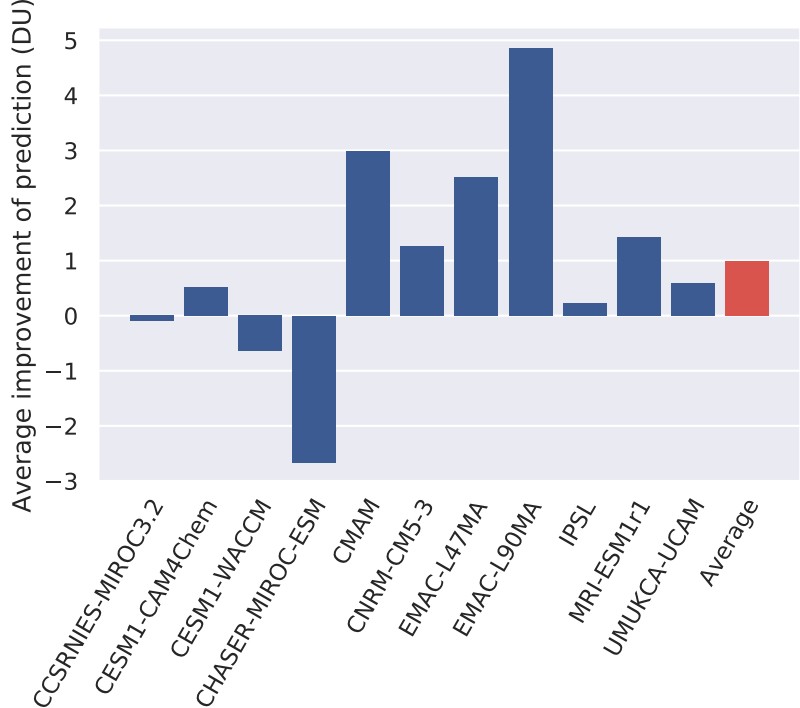

**Figure 4.** Results of the perfect model test. The average improvement in the Antarctic October TCO projection (1960–2095) of the WM compared to the MMM for each model taken as the pseudo truth. The average shown in red is the improvement across all the perfect model tests. No conclusions about overall model skill should be drawn from this plot.

## 4.3 Model independence

The current design of model inter-comparison projects does not account for structural similarities in models, ranging from
sharing transport schemes to entire model components. Therefore, a key part of generating an informed weighting is considering how alike any two models are. The weighting scheme presented here accounts for model independence through the denominator in Eq. (1).

The refC1SD scenario from CCMI consists of 14 different simulations, some of which are with different models, whereas others are just different realisations of the same models. Note that there are more models used here than in the creation of the
Antarctic ozone projection. This is because for the weighted projection we require both a refC1SD and a refC2 simulation for each model, but for similarity analysis we can use all the refC1SD simulations. For these model runs we calculated a similarity





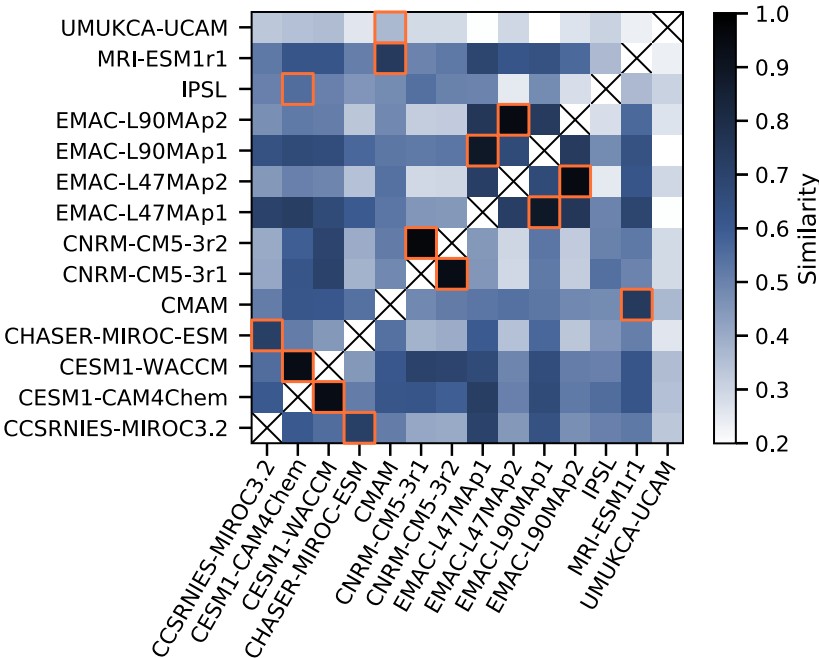

**Figure 5.** Inter-model similarity across all refC1SD models as calculated by Eq. (3). A similarity of 1 denotes models which are identical for all the metrics, whereas a lower similarity shows a greater independence. The orange boxes highlight the model most similar to the model on the y-axis.

index $s_{ij}$ (shown in Eq. (3)) which is the similarity between models $i$ and $j$ averaged across all the performance metrics, where $n_k$ is the size of the data for metric $k$.

$$s_{ij} = \frac{1}{M} \sum_{k=1}^{M} \exp\left( \frac{-S_{ijk}^2}{\sigma_S n_k} \right) \tag{3}$$

Similarities between all refC1SD models are shown in Fig. 5. We also found the maximum value of $s_{ij}$ for each model, indicating the model which model $i$ is most similar to. The most alike models are the two realisations of CNRM-CM5-3, which are the same models running with slightly different initial conditions. We also see high similarity between the two variations of the CESM model, CESM-WACCM and CESM-CAM4Chem. CESM1-CAM4Chem is the low-top version of CESM1-WACCM, meaning that up to the stratosphere the two models should be much alike (Morgenstern et al., 2017).

Analysing the EMAC models this way presents an interesting observation: changing the nudging method, has a greater impact on model similarity than changing the number of vertical levels (the difference between EMAC-L47MAr1i1p1 and EMAC-L47MAr1i1p2, and likewise the 90 level model variant, is that the p1 variant additionally nudges to the global mean temperature (Jöckel et al., 2016)). CHASER-MIROC-ESM and CCSRNIES-MIROC3.2 are two other models which are identified as similar albeit at a lower value. Considering that these two models are built upon the same MIROC general circulation model it is not





a surprise that we see a similarity. That the weighting framework can identify all of the models with known similarities (same institution, or realisations) confirms confidence in the methodology and means that we are down-weighting similar models.

## 5    Discussion

The projection of the ozone hole recovery date presented here makes use of an ensemble of the latest generation of CCMs and a weighting methodology that accounts for complexities within model ensembles. While the ozone recovery date found in this

work (2056) is different to that found by Dhomse et al. (2018) (2062), these two dates are not easily comparable as they are created from different subsets of the same ensemble. For our subset of models, the MMM recovery date was 3 years earlier (2053) than the WM. Although the return dates are not significantly different, for the period of peak ozone depletion (especially between 1990 and 2030) the MMM projection is significantly different to the WM. As the model subsets in this work, for the WM and MMM remain the same, the variation in the projections is entirely due to the construction of the WM.

The CNRM-CM5-3 model received the largest weight of $0.27$, giving it three times the influence in the WM than in a MMM. Initially this may seem as if we are placing too much importance on one model, but consider that in a standard MMM, a model which runs three simulations will have three times the influence of a model with a single simulation. Furthermore, CNRM-CM5-3 is not weighted higher because it ran more simulations, it is weighted higher because it is skilful at simulating hindcasts whilst maintaining a level of independence.

Central to the weighting methodology is the selection of metrics requiring expert knowledge. The set of metrics we chose, were grounded in scientific understanding and produce a good improvement of the weighted projection compared to the MMM. There are numerous other metrics of varying complexity which could be considered, such as the size of the ozone hole or the abundance of polar stratospheric clouds. These extra metrics could improve the model weighting and give a more accurate projection, but testing an exhaustive collection of metrics was not our aim, and there are not always appropriate measurements

to validate the metrics with. We have shown a weighting framework which improves upon the current methodology for combining model ensembles, and is also flexible and adaptable to which ever metric choices the user deems reasonable. Furthermore, the low range in return dates produced from the dropout testing shows that the results produced in this weighting framework are robust to metric and model choices. This is a desirable effect of a methodology to provide stable results irrespective of fluctuations in the input.

It is reassuring to know that the methodology is robust to metric choices as we are often constrained by the availability of observational data. In this work we benefit from the decades of interest in polar ozone which have led to datasets of a length suitable for constructing model weights. This highlights the importance of continued production of good observational datasets because, although perfect model testing allows us a form of testing which forgoes the need for observations, weighting methodologies must be grounded in some estimate of the truth.

Abramowitz et al. (2019) discuss approaches for assessing model dependence and performance, and mention caveats around the notion of temporal transitivity: is model behaviour comparable between two distinct temporal regions? Here, we rephrase the question to be: are the weights generated from the hindcast scenario relevant and applicable to the forecast scenario? This





not only questions temporal transitivity, but also that models may have codified differences between scenarios in addition to differences in physical and chemical regimes. In this study, scenario transitivity (as we call it) is demonstrated through perfect

model testing. On average the WM produced a better (closer to the pseudo truth) projection than if we had considered the MMM. This shows that weights calculated from the refC1SD hindcasts produce better projections from the refC2 forecasts and are therefore transitive between the two scenarios.

We generated weights from the refC1SD simulations which means that some metrics we chose are based on nudged variables, such as the lower stratospheric temperature gradient. As a result, one might expect that the model skill for these metrics should

be equal, though given Fig. 3 this is not true. One may then expect that the weighting is not capturing model skill, but instead the skill of the models' nudging mechanisms. This is harder to test, with such a variety in nudging time-scales and methods. We justified the use of using the nudged refC1SD simulations, despite these considerations, for two reasons. Firstly, that nudged simulations give the models the best chance at matching the observational record, by providing relatively consistent meteorology across the models. The free running CCMI hindcast simulations (refC1) have a large ensemble variance and,

despite producing potentially realistic atmospheric states, are not directly comparable to observational records. Secondly, the perfect model testing discussed above, demonstrates that the nudging doesn't have a negative effect on the weighting. As the perfect model test produces better projections, for models which are nudged in a variety of ways, we can conclude that the weighting is not measuring nudging. Take for example UMUKCA-UCAM which is nudged quite differently, as evidenced by a colder pole than the ensemble. If the methodology was testing nudging, we would expect the perfect model test, when using

UMUKCA-UCAM as the pseudo truth, to not produce a WM projection which was better than the MMM, because the nudging in UMUKCA-UCAM is not like any other model in the ensemble. However, this is not the case, and the WM projection is better than the MMM, confirming that the weighting is not largely dependent on the nudging process.

Although we were not seeking to grade the CCMs as per Waugh and Eyring (2008), the construction of a weighted mean provides insight into model performance which would not be considered in a MMM. This is of some relevance as the CCMI

ensemble has not undergone the same validation as its predecessors, such as CCMVal (Eyring et al., 2008). Additionally, we gain insight into model dependence shown in Sect. 4.2. Whilst this approach may not be as illuminating as Knutti et al. (2013), where they explored the genealogy of CMIP5 models through statistical methods, or Boé (2018), who analysed similarity through model components and version numbers, it successfully identified the known inter-model similarities. More complex methods are desirable, especially those that consider the history of the models' developments. Nevertheless, the simplicity of

quantifying inter-model distances as a measure of dependence lends itself well to model weighting.

## 6  Conclusions

We have presented a model weighting methodology, which considers model dependence and model skill. We applied this over a suite of metrics grounded in scientific understanding to Antarctic ozone depletion and subsequent recovery. In particular we have shown that the weighted projection predicts recovery by 2056 with a 95 % confidence interval of 2052–2060. Through



perfect model testing we demonstrated that on average a weighted mean performs better than the current community standard of calculating a multi model mean.

This methodology addresses the known shortcomings of an ensemble multi model mean which include, the problem of ensembles including many similar models, and the inability to factor in model performance. It does this by quantifying skill and independence for all models in the ensemble over a selection of metrics which are chosen for their physical relevance to

the phenomena of interest. This weighting methodology is still subject to some of the same limitations of taking an ensemble mean: i.e., we are still limited by what the models simulate. For example, in the case of ozone depletion, a weighted mean is no more likely to capture the ozone changes due to the recent fugitive CFC-11 release (Rigby et al., 2019). Instead it allows us to maximise the utility of the ensemble and, provided we are cautious of over-fitting, it allows us to make better projections.

Addressing the shortcomings and presenting possible improvements of methods for averaging model ensembles is timely

given the current running of CMIP6 simulations (Eyring et al., 2016). That ensemble could arguably be the largest climate model ensemble created to date, in terms of the breadth of models considered. Therefore, the need for tools to analyse vast swathes of data efficiently for multiple interests is still growing. The models within CMIP6 are likely not all independent, which could affect the robustness of results from the ensemble by biasing the output towards groups of similar models. The similarity analysis within this work would allow users of the ensemble data to understand if ensemble biases are emerging from

similar models and acknowledge how this may impact their results.

In summary, we have presented a flexible and useful methodology, which has applications throughout the environmental sciences. It is not a silver bullet for creating the perfect projection for all circumstances; however, it can be used to construct a phenomenon-specific analysis process that can account for model skill and model independence, both of which will improve ensemble projections compared to a multi-model mean.



*Code and data availability.* The jupyter notebook used to run the analysis, along with a collection of functions to produce weightings from ensembles, can be found at https://doi.org/10.5281/zenodo.3624522. The CCMI model output was retrieved from the Centre for Environmental Data Analysis (CEDA), the Natural Environment Research Council's Data Repository for Atmospheric Science and Earth Observation (http://data.ceda.ac.uk/badc/wcrp-ccmi/data/CCMI-1/output), and from NCAR's Climate Data Gateway (http://www.earthsystemgrid.org).

*Author contributions.* MA developed the methods and led the analysis, and conceived the study alongside JSH and PJY, who made major contributions as the work progressed. MA drafted the manuscript, with the guidance of PJY and JSH and input from JFL. JFL and all the other co-authors provided model simulation data. All the co-authors provided model output and helped with finalising the manuscript.

*Competing interests.* The authors declare that they have no conflict of interest.

*Acknowledgements.* This work was supported by the Natural Environment Research Council [NERC grant reference number NE/L002604/1], with Matt Amos's studentship through the ENVISION Doctoral Training Partnership. Paul J Young is partially supported by the Data Science of the Natural Environment (DSNE) project, funded by the UK Engineering and Physical Sciences Research Council (EPSRC; grant number EP/R01860X/1). We would like to thank Bodeker Scientific, funded by the New Zealand Deep South National Science Challenge, for providing the combined NIWA-BS total column ozone database. We also acknowledge the GOZCARDs team for the production of the HCl record and Remote Sensing Systems for the MSU TLS record. We acknowledge the modelling groups for making their simulations available for this analysis, the joint WCRP SPARC/IGAC Chemistry-Climate Model Initiative (CCMI) for organising and coordinating the model data analysis activity, and the British Atmospheric Data Centre (BADC) for collecting and archiving the CCMI model output. The EMAC simulations have been performed at the German Climate Computing Centre (DKRZ) through support from the Bundesministerium für Bildung und Forschung (BMBF). DKRZ and its scientific steering committee are gratefully acknowledged for providing the HPC and data archiving resources for this consortial project ESCiMo (Earth System Chemistry integrated Modelling). The CCSRNIES-MIROC3.2 model computations were performed on NEC-SX9/A(ECO) and NEC SX-ACE computers at the CGER, NIES, and supported by the Environment Research and Technology Development Funds of the Ministry of the Environment, Japan (2-1303) and Environment Restoration and Conservation Agency, Japan (2-1709).




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
