# Peer review of "chemistry-climate models weighted by model performance and"

_Atmospheric Chemistry and Physics, 2020_

## Referee Comment (RC1) · Anonymous Referee #1 · 10 Mar 2020

Review of 'Projecting ozone hole recovery using an ensemble of chemistry-climate models weighted by model performance and independence' by M. Amos et al.

The study by M. Amos and colleagues looks into the historical and future evolution of the Antarctic ozone hole using a weighting technique which accounts for model independence and performance. It is based on an ensemble of Chemisty-Climate Models, which are combined into a weighted multi-model mean by applying a weighting method established in the literature. The authors find that ozone returns to 1980 levels about 3 years later in the weighted mean compared to the unweighted one.

The manuscript is very well written and concise which makes it easy to follow the au-

thors arguments. In addition, the authors openly address several potential shortcomings of their approach and included an extensive analysis of the performance based on cross-validation, which I find to be an excellent example of best practice. While the results the authors find are not ground-braking, the work, in my opinion, still presents a valuable contribution to the field in particular in highlighting the importance of accounting for model inter-dependence when calculating uncertainty intervals based on MME spread. I found some unclear statements which should be addressed as noted in my minor comments below otherwise I recommend this contribution for publication in ACP.

Minor comments

L1: I don't think that there is a single, agreed-on 'current method' for averaging model ensembles in climate science.

L70: This is a minor point but I'm just pointing out that the REA does not down-weight models if they are more similar (dependent). It rather does the opposite and gives models which are closer to the multi-model mean additional weight as they are considered to be more reliable.

L90: There are several more recent papers addressing and further developing this weighting method: http://doi.wiley.com/10.1029/2017JD027992 https://doi.org/10.1088/1748-9326/ab492f

Equation 1: The sum in the denominator should run over j!=i I assume?

Section 2.1/Figure 1: I fail to understand how the sigma values where chosen from the figure and the text in this section. Can the authors elaborate on that please?

L231: delete 'trend'? Figure 2 shows a time series of TCO not a trend right?

L252: Figure 2 → Figure 3

l268: Maybe give the weight here relative to equal weighting similar to the lowest weight later in the paragraph. This seems to be the more meaningful metric as the absolute

weight depends on the ensemble size.

Figure 4: This might just be an ambiguity in the language, so let me see if I understand correctly: Given a perfect model X, there is an unweighted projection Y and a weighted projection Y' correct? So what is shown here is the difference abs(X-Y)-abs(X-Y')? If that is they case maybe rather call it the improvement of the (weighted) average than the average improvement? (except the last bar). I was a bit confused about the "Average improvement" which seems to indicate that this is showing some kind of average over different improvements (if my interpretation is correct).

L296: 'alike any two models are' I don't think that this should be the aim or at least it should be worded more carefully. Two models can be 'alike' for different reasons, one of them being that they simulate the same system and are both 'good' at it so that the converge towards the truth. Or in other words: models should not be punished for arriving at the same answer independently. The assumption behind independence weighting schemes is that an inference on the model dependence can be made based on their output. If, e.g., two models share biases in several metrics this could be a sign that they also share one or several components.

L355: Have the authors tried to create the weights based on the refC2 simulations directly? Is it possible to receive sensible weights from that and how do they compare to the weights retrieved from the hindcasts?

---

## Referee Comment (RC2) · Anonymous Referee #2 · 11 Mar 2020

"Projecting ozone hole recovery using an ensemble ofchemistry-climate models weighted by model performance and independence"- Review

Authors use Chemistry Climate Model (CCMI) data to estimate Antarctic ozone recovery. They use newer weighting methodology to investigate evolution of Antarctic ozone depletion and subsequent recovery from 11 CCMs. Estimated ozone recovery dates are 2056 (2052–2060), that is about 4 years earlier than Dhomse et al., 2018. Matrices used to construct weighted means are total ozone column gradient, lower stratospheric temperature, lower stratospheric temperature gradient, breakdown of the polar vortex, ozone-temperature gradient, ozone trend- temperature trend gradient, and hydrogen

chloride.

Overall, this is well written concise paper and I think this is somewhat revised version of the manuscript. I would recommend it for a publication if authors can incorporate some of the following comments.

Major comment: Line 359 "The free running CCMI hindcast simulations (refC1) have a large...."

A) Which should be true for refC2 simulation as well, hence estimated ozone recovery dates should have much large uncertainty. I think authors should give some clearer and better explanation for the selection of refC1SD over refC1 or first part of refC2 to calculate the weights. It is odd that weights are calculated for completely different dynamical space as ozone evolution would largely determined by the changes in the stratospheric dynamics. For me higher weights to CNRM model over WACCM is really odd. Simple October TCO time series comparison (as well as ozone profile comparison) suggests CNRM being bit outlier. So I am wondering high weightage to CNRM might be due to stronger nudging parameters. I think authors tried to explain in the paragraph starting at 353, but it is confusing and better explanation would help the readers.

B)Section 3.2: Please provide some more details about which pressure levels are used for lower stratospheric temperature, ozone. Also what does HCl averaged over whole stratosphere means? That does not make sense. Do you convert it in number density and calculate stratospheric column. Chlorine activation in the lower most stratosphere determines springtime ozone loss and mid-stratospheric or upper stratospheric HCl values are not that important.

Minor comments: i) Line 25: [e.g. Gillet, 2015, ..) ii) Line 34: Ball et al., 2018 is not really good reference for that sentence. iii) Table 2. Reference NIWA data V3.4 should be Bodeker et al., (2018) url = {https://doi.org/10.5281/zenodo.1346424} iv) line 327: that is not correct. In MMM, if model has more than one realization then genrally individual model time series is created by calculating ensemble mean. If there is only

one realization then mot of the studies use 3 box-smoothing window.

---

## Referee Comment (RC3) · Anonymous Referee #3 · 19 Mar 2020

General Comments

The manuscript by Amos et al. introduces a method aimed at creating an alternative to the multi-model mean when examining an ensemble of model simulations. The weighting technique presented here accounts for both model performance and model similarity, with higher weightings given to models that compare well to observations and that are more independent of other models in the ensemble. The method is demonstrated using predictions of Antarctic ozone hole recovery from the future simulation of the Chemistry-Climate Model Initiative. The demonstration notably incorporates several observational metrics of species/physical parameters relevant for Antarctic strato-

spheric ozone.

A need for this type of analysis exists; it has been conveyed from many in the global modeling community that a simple multi-model mean, where all models are equally weighted, is not sufficient. Therefore, I consider the subject of the manuscript important, and the scientific findings presented through the demonstration of the method are relevant for ACP. I have some broader questions regarding the general applicability of the technique, but view the paper overall to be well-written, appropriate for the journal, and ready for publication after the authors have addressed the following minor comments. It is also a fitting submission for the CCMI special issue.

Specific Comments

L. 148 and throughout: What about when poor model performance manifests as poor simulation of the dynamics? If a model has an accurate chemical mechanism, maybe it looks good in the SD simulations, but it's poor when simulating the Antarctic vortex in free-running mode – doesn't that mean it should not be trusted to get the future ozone hole recovery right?

L. 194: How are the temperature metrics influenced by the SD versus free-running simulations? What would happen if you calculated the individual and total weights (i.e., Fig. 3) using Ref-C2 instead of Ref-C1SD? And, similar to the previous comment, what if the nudging in the SD run is what causes a realistic decrease in temperature, not the coupling between decreased ozone and temperature (i.e., if ozone is poorly simulated, but the nudging imposes realistic temperature changes, will a high weighting be awarded to this model, for this metric, despite getting temperature "right for the wrong reasons")?

L. 205: At this point, it's not clear if each of these metrics will be tested individually, or if all of them will be combined as in Eq. 2, or various combinations will be tested? Later in the manuscript, a dropout test is described; perhaps that should be moved earlier (under "3.3 Evaluating the weighting framework") to place it in the larger context

sooner?

L. 290: Are the results of this out-of-sample test shown anywhere? It's difficult for me to grasp what these RMSE values mean, in context, though I'd be curious to see the results of the test.

L. 336: On the sensitivity of the final weightings to which performance metrics are included: You performed a dropout test, leaving one metric out at a time. But, what about leaving out two? For instance, since CNRM-CM5-3 stands out so significantly in its Total Weight, what if you leave out the two metrics where that model apparently excels above the other models in performance, i.e., Polar vortex breakdown trend and Ozone-temperature gradient? I am concerned that the Total Weight for each model is highly sensitive to the combination of observational metrics included, beyond what is tested by the single-metric dropout test.

Technical Corrections

L. 50: "…and climate" should be "and climate forcers" or similar?

L. 200: "high" should be "highly" and "except of" should be "except for"

L. 252: Should "Figure 2" instead be "Figure 3"?

L. 267: "The total weighting, formed from the summation of individual metric weights…" Is this right? This conveys to me that all of the individual metric weights are simply totaled. Since CNRM has a couple weights >0.4, then its resulting total weight of 0.27 suggests this is not right… Found by Eq. 2 instead?

L. 273: "The lowest model weight is 55 % the value of a uniform weighting." I understand this to mean that the lowest red bar in Fig. 3 is 55% the magnitude of the dashed black line, but the smallest bars (CCSRNIES-MIROC3.2 and EMAC-L47MA) look to be less than half, judging by eye. Is this statement correct?

---

## Author Response (AR1)

We would like to thank the 3 reviewers and the editor for their time in considering the paper and for their constructive comments. Our responses to these comments (repeated in blue) are below.

**Response to reviewer 1**

The manuscript is very well written and concise which makes it easy to follow the authors arguments. In addition, the authors openly address several potential shortcomings of their approach and included an extensive analysis of the performance based on cross-validation, which I find to be an excellent example of best practice.

Thank you for such a kind comment!

L1: I don't think that there is a single, agreed-on 'current method' for averaging model ensembles in climate science.

This is a fair point to make. We have changed the first sentence to highlight that standard averaging is commonly used, whilst not implying that it is the community standard.

Was:
The current method for averaging model ensembles, which is to calculate a multi model mean, assumes model independence and equal model skill.

Changed to:
Calculating a multi model mean, a commonly used method for ensemble averaging, assumes model independence and equal model skill.

L70: This is a minor point but I'm just pointing out that the REA does not down-weight models if they are more similar (dependent). It rather does the opposite and gives models which are closer to the multi-model mean additional weight as they are considered to be more reliable.

We appreciate the clarification. We have changed the sentence to reflect what you have said. It now reads:

Additionally, reliability ensemble averaging (REA) (Giorgi and Mearns, 2002) is an alternative weighting technique which instead gives higher weights to those models near the multi model mean.

L90: There are several more recent papers addressing and further developing this weighting method: http://doi.wiley.com/10.1029/2017JD027992 https://doi.org/10.1088/1748-9326/ab492f

Thank you for the heads up, these are now included.

Equation 1: The sum in the denominator should run over j!=i I assume?

Yes! Thank you for spotting that.

Section 2.1/Figure 1: I fail to understand how the sigma values where chosen from the figure and the text in this section. Can the authors elaborate on that please?

The purpose of Figure 1 is to illustrate how the weight depends on both the independence and the performance for a given value of sigma_d and sigma_s. These are not the sigma values used in the paper, it is instead a toy data set.

We deliberately didn't include too much description on the method of determining the sigma values, as we appreciate that there isn't a particularly objective way of doing so, as noted in by Knutti et al. (2017). In our case, to determine the values, we treated it somewhat like a machine learning problem, by having training and testing sets of data which don't overlap. The training set in this case was the refC1SD data including the total ozone column projection. The sigma values were found by optimising the weights, such that when they were applied to the refC1SD total ozone column they were a good fit to the observations. The testing set is then the weights applied to refC2 projections, which can be tested temporally out of sample (2010-2016), which avoids performing testing on data that has already been seen.

There may well be alternative values of sigma that deliver different fits to the observations and different scores in a perfect model test. But we believe that these sigma values balance performance and independence, and are found in a fair way.

The above information is reflected in a new short paragraph at the end of section 2.1 (from Line 126 in the new manuscript) that clarifies the choosing of the values of sigma.

L231: delete 'trend'? Figure 2 shows a time series of TCO not a trend right?

It is a trend in the sense that it is a smoothed time series with the inter-annual variability removed. The process for creating the trend is described in the paragraph beginning line 163. The term used by Scinocca et al. (2010) (the TSAM paper) is 'individual model trend' and for continuity we prefer to use that terminology.

L252: Figure 2→Figure 3

Indeed. Thanks!

L268: Maybe give the weight here relative to equal weighting similar to the lowest weight later in the paragraph. This seems to be the more meaningful metric as the absolute weight depends on the ensemble size.

Yes, we think that would add clarity, and possibly reassure readers that a weighting of 0.27 is not actually that large. The bold section has been added:

…CNRM-CM5-3, which has a weight of 0.27 **(297% the value of a uniform weighting.)**

Figure 4: This might just be an ambiguity in the language, so let me see if I understand correctly: Given a perfect model X, there is an unweighted projection Y and a weighted projection Y'

correct? So what is shown here is the difference abs(X-Y)-abs(X-Y')? If that is they case maybe rather call it the improvement of the (weighted) average than the average improvement? (except the last bar). I was a bit confused about the "Average improvement" which seems to indicate that this is showing some kind of average over different improvements (if my interpretation is correct).

The average here ('The average improvement in the Antarctic October TCO projection') is used because it is an average over the time series. You are correct, the improvement is the abs(multi model mean – pseudo truth) - abs(weighted mean – pseudo truth). We do see your point about ambiguity. To clarify that the average is temporal, we have changed 'average improvement' to '**mean monthly** improvement', in both the text and the y axis label of figure 4.

L296: 'alike any two models are' I don't think that this should be the aim or at least it should be worded more carefully. Two models can be 'alike' for different reasons, one of them being that they simulate the same system and are both 'good' at it so that the converge towards the truth. Or in other words: models should not be punished for arriving at the same answer independently. The assumption behind independence weighting schemes is that an inference on the model dependence can be made based on their output. If, e.g., two models share biases in several metrics this could be a sign that they also share one or several components.

Although, as you say, models shouldn't be punished for reaching the right answer independently, we should be more cautious of models which generate the right ozone column output but fail to simulate well any of the processes that are important in simulating ozone (e.g. temperature). That is one of the underlying reasons for analysing the models output this way: we want to have confidence that the models are simulating things for the right reason. This is important because we want to extend the weights onto the forecast scenario (refC2). By knowing that the models are simulating the ozone concentration well, because they simulate the underlying physics and chemistry well, we have more confidence that they will do so in the forecast.

Of course, if two very similar models both perform very well, we can see that they both simulate well for the right reason, because this will be reflected in the performance metrics. To some extent this can be seen in the temperature weighting of UMUKCA (Fig 3). UMUKCA has a pole significantly colder than the observations (and the other models) and resultantly gets heavily down weighted. It does not however receive a significant up weighting due to it being very different from the rest of the models.

Additionally, we have the ability to tune the sigma parameters. If we had a selection of near perfect models, the emphasis can be put on the performance aspect instead of the independence aspect.

L355: Have the authors tried to create the weights based on the refC2 simulations directly? Is it possible to receive sensible weights from that and how do they compare to the weights retrieved from the hindcasts?

Throughout the conception and refinement of this work we've had numerous discussions about what is the fairest set of simulations to create the weights from, and we hope we've highlighted the important parts of these discussions in the text.

To be specific, we have constructed weightings from both the refC1 simulations (essentially free running hindcasts) and refC2. In our opinion the weights from these were not sensible for a number of reasons:
- The refC1 set of simulations is not large and not many models ran multiple realisations. As a result, we don't have a good cover over the possible variable space, meaning that only a couple of models received a strong performance weighting. The same issue was true for the refC2 simulations. This free running non-smoothed set up also meant that the dependence weightings are particularly useful because of the large range of model output, for example the range of the total column ozone values can be as large as 250DU which is almost 100% of the 'normal' ozone column.
- To remove the interannual variability, to allow for a fair comparison, we would have to smooth metrics to allow for a trend. Missing data makes the smoothing tricky and means that we lose an amount of data.
- Essentially, we wanted to give the models the best chance of replicating the observations and the way to do this was to use the specified dynamics runs. This allows us to use all the available observational data and to test model response to events which would be lost through smoothing, such as volcanic eruptions and sudden stratospheric warming events. There is a brief discussion of this from L379 onwards.

**Response to reviewer 2**

Overall, this is well written concise paper and I think this is somewhat revised version of the manuscript.

Thank you. We very much appreciate your kind comments.

Major comment: Line 359 "The free running CCMI hindcast simulations (refC1) have a large...."

We've broken up your comments, so they can be addressed individually.

A) Which should be true for refC2 simulation as well, hence estimated ozone recovery dates should have much large uncertainty.

We have very carefully followed the work of Scinocca et al. (https://agupubs.onlinelibrary.wiley.com/doi/full/10.1029/2009JD013622) when calculating the model trends and their associated uncertainties. This is a popular way for robustly extracting model trends whilst learning ensemble uncertainties. It is especially fitting for use here, given the methodology accounts for individual model weights. Having double checked the implementation of this method, we do believe that the uncertainties we calculate are genuine.

We clarify here that the confidence interval of the trend is for the trends with all interannual variability removed. The uncertainty presented by the confidence interval, is not a measure of the spread of the models in the ensemble. Instead it is a measure of the confidence in each of the individual model trends. For this reason, looking at the spread of the refC2 ensemble is not a good indicator of what the confidence interval of the multi model trend will be.

I think authors should give some clearer and better explanation for the selection of refC1SD over refC1 or first part of refC2 to calculate the weights. It is odd that weights are calculated for completely different dynamical space as ozone evolution would largely be determined by the changes in the stratospheric dynamics.

With reference to the metrics representing the relevant dynamical space, we believe the weights are generated from an appropriate set of metrics, which are relevant to ozone evolution the reasons for which are set out in section 3.2. With explicit reference to stratospheric dynamics, one of the metrics we include is based on the polar vortex which is extremely influential in southern polar dynamics in and around wintertime. We do highlight as well, that a better set of metrics could be chosen, but we are constrained by observational availability and model output.

To justify the use of the weights generated using refC1SD on the refC2 simulations we performed both an out of sample test and a perfect model test (detailed in section 4.2). This shows that the weights learnt in one dynamical space are applicable to a different dynamical space. We also note that recent work from Orbe et al., (2020) that model biases that exist in a free running model can also exist in its specified dynamics counterpart (it is also true that it is partly down to the implementation of the specified dynamics). This would indicate that there is greater similarity between the dynamical spaces than you suggest, and therefore reason to use refC1SD for the selection weights.

Here, we repeat our response to a similar comment from the first reviewer, to further address the use of the refC1SD simulations. We did additionally construct weightings from both the refC1 simulations (essentially free running hindcasts) and refC2. In our opinion the weights from these would not be sensible for a number of reasons:
- The refC1 set of simulations is not large and not many models ran multiple realisations. As a result, we don't have a good cover over the possible variable space, meaning that only a couple of models received a strong performance weighting. The same issue was true for the refC2 simulations. This free running non-smoothed set up also meant that the dependence weightings are particularly useful because of the large range of model output, for example the range of the total column ozone values can be as large as 250DU which is almost 100% of the 'normal' ozone column.
- To remove the interannual variability, to allow for a fair comparison, we would have to smooth metrics to allow for a trend. Missing data makes the smoothing tricky and means that we lose an amount of data.
- Essentially, we wanted to give the models the best chance of replicating the observations and the way to do this was to use the specified dynamics runs. This allows us to use all the available observational data and to test model response to events which would be lost through smoothing, such as volcanic eruptions and sudden stratospheric warming events.

For me higher weights to CNRM model over WACCM is really odd. Simple October TCO time series comparison (as well as ozone profile comparison) suggests CNRM being bit outlier. So, I am wondering high weightage to CNRM might be due to stronger nudging parameters. I think authors tried to explain in the paragraph starting at 353, but it is confusing and better explanation would help the readers.

Yes, CNRM does appear to be an outlier in the ensemble, but when we look at where the observations fit in the model spread (of CCMI refC1SD models) we see that the CNRM model is actually one of the closest to the observations (plot below). If the weights, as you suggest, are representative of higher nudging then we are up weighting the model which comes closest to the observations, which is a good thing.  A way to test whether the nudging is having an effect on the weighting is to use the perfect model test.

[Figure]

In response to this comment, we have rewritten a large part of section 5 (L359-378) to clarify the utility of the perfect model test and why it shows that using nudged models doesn't have a detrimental effect on model weighting. The new text is below.

We generated weights from the refC1SD simulations which means that some metrics we chose are based on nudged variables, such as the lower stratospheric temperature gradient. As a result, one might expect that the model skill for these metrics should be equal, though given Fig. 3 this is not true. One may then expect that the weighting is not capturing model skill, but instead the skill of the models' nudging mechanisms; the models are nudged on different timescales ranging from 0.5 hr to 50 hr and from varying reanalysis products (Orbe et al., 2020). We use the perfect model test to show that the utility of the weighting methodology is not compromised by using models with such a variety in nudging time-scales and methods.  As the perfect model test produces better projections, for models which are nudged in a variety of ways, we can conclude that the weighting is not dominated by nudging. Take for example UMUKCA-UCAM which is nudged quite differently compared to the ensemble, as evidenced by a southern pole significantly colder than the ensemble. When we take UMUKCA-UCAM as the pseudo truth (temporarily assuming the UMUKCA-UCAM output is the observational truth) we generate weights based

upon the refC1SD simulations and test them on the refC2 simulations. The weights generated are based on the dynamical system simulated in refC1SD which includes any model nudging. We can test how well these weights apply to a different dynamical system without nudging (refC2). The improvement in the WM compared to the MMM suggest that the weights generated from the refC1SD dynamical system do not predominantly reflect the quality of nudging and can be applied. If there hadn't been an improvement, then the dynamical systems described by refC1SD and refC2 may be too dissimilar for this weighting methodology and the weights may instead have been dominated by how well models are nudged. Nudging may be influencing the weights, but not to a degree that the accuracy of the projection suffers. Orbe et al. (2020) highlight the need for care when using the nudged simulations and we would like any future work on model weighting to quantify the impact of nudging upon model weights to reflect this.

We justified using the nudged refC1SD simulations, despite these considerations, for two reasons. Firstly, these nudged simulations give the models the best chance at matching the observational record, by providing relatively consistent meteorology across the models. The free running CCMI hindcast simulations (refC1) have a large ensemble variance and, despite producing potentially realistic atmospheric states, are not directly comparable to observational records. Secondly, the perfect model testing discussed above, demonstrates that the nudging doesn't have a detrimental effect on the model weighting.

B) Section 3.2: Please provide some more details about which pressure levels are used for lower stratospheric temperature, ozone.

- The ozone used for the projections (Fig 2) is used from the toz (total ozone) variable that the modelling groups output; we do not construct that.
- The temperature is taken as a weighted average over the lower stratosphere using the MSU TLS weighting function (i.e., a non-uniform average over a range of pressures). We have added a citation to the weighting function in section 3.2, in addition to the original citation in table 2. All further uses of temperature in any metric are constructed this way also. This is as described by Mears and Wentz (2009).
- The ozone used in the metrics is again total column ozone direct from the model output. The models do have different vertical ranges as discussed in section 3.1. We appreciate that we may have excluded some details readers were interested in. In section 3.2, any mention of 'ozone' has been changed to 'total column ozone'

Also, what does HCl averaged over whole stratosphere means? That does not make sense. Do you convert it in number density and calculate stratospheric column? Chlorine activation in the lower most stratosphere determines springtime ozone loss and mid-stratospheric or upper stratospheric HCl values are not that important.

Agreed, this is quite vague terminology. The HCl, as it is taken from the GOZCARDS product, has areas (both in vertical levels and latitude) that have missing data. So, we take all the model HCl output and regrid it to the same vertical and horizontal coordinates of the original HCl observations. These are constrained over a polar cap (90°S, 65°S) from levels between 316hPa and 15hPa (9 levels). For each spring season (SON) we take all the available data and create an average HCl concentration between 316hPa and 15hPa, from 90°S to 65°S. This is done in such a way that any data missing in the observations is excluded in the model data.

We have updated the hydrogen chloride part in Section 3.2 for clarity to add:

We consider a pressure range of 316 hPa to 15 hPa to capture the concentration in the lower stratosphere.

Minor comments:
i) Line 25: [e.g. Gillet, 2015, ..)

Good spot, thanks!

ii) Line 34: Ball et al., 2018 is not really good reference for that sentence.

Yes, fair point – reference removed.

iii) Table 2. Reference NIWA data V3.4 should be Bodeker et al., (2018) url={https://doi.org/10.5281/zenodo.1346424}

Thank you.

iv) line 327: that is not correct. In MMM, if model has more than one realization then generally individual model time series is created by calculating ensemble mean. If there is only one realization then most of the studies use 3 box-smoothing window.

True, though a naïve mean would average across all simulations. We have added the bold section below to reflect, say, HadGEM in CMIP5 having HadGEM2-AO, HadGEM2-CC and HadGEM2-ES:

Initially this may seem as if we are placing too much importance on one model, but consider that in a standard MMM, a model which runs three simulations **with different combinations of components** will have three times the influence of a model with a single simulation.

**Response to reviewer 3**

…but view the paper overall to be well-written, appropriate for the journal, and ready for publication after the authors have addressed the following minor comments.

Thank you!

L. 148 and throughout: What about when poor model performance manifests as poor simulation of the dynamics? If a model has an accurate chemical mechanism, maybe it looks good in the SD simulations, but it's poor when simulating the Antarctic vortex in free-running mode – doesn't that mean it should not be trusted to get the future ozone hole recovery right?

This is a very good question and raises a similar point to your next comment. Overall, in line with this comment and one from a different reviewer we have rewritten a large part of section 5

(L359-378) to clarify the utility of the perfect model test and why it shows that using nudged models doesn't have a detrimental effect on model weighting. A detailed response follows below.

Your question is essentially one about the nudging in the specified dynamic runs and whether this is influencing the measure of model performance. We have addressed this through the use of the perfect model test. The perfect model test pretends that the output from one model (in turn) is the "observational truth" (pseudo truth). This gives us the ability to have a full set of data both for the past (refC1SD) and the future (refC2). If what you're saying is true, that a model is only good in the past because it is nudged, then the perfect model test should show us this. For example, if we have a model which is strongly nudged in refC1SD so that it gets highly weighted because it is a good fit for the pseudo truth, when we test the fit of the weighted mean to the refC2 projections we will see that the weighted mean is actually a poor fit. In this case the perfect model test will show us that the weighted mean is not a good predictor of the future because the dynamical systems between refC1SD and refC2 are very different. However, if the weighted mean is a good predictor of the future (we measure compared to the multi-model mean), then we conclude that there is sufficient overlap (of the dynamical space) between the model in refC1 and refC2, and that the nudging hasn't had a large detrimental impact. The results of the perfect model test show that a weighted mean, constructed from refC1SD simulations, is a better projection for refC2 than a multi model mean and therefore, simulation performance is at least partly transitive between refC1SD and refC2. Additionally, we perform a small out of sample test with the overlap between observations and the beginning of the refC2 simulations (L287) which shows that the weights from the specified dynamics simulation, when applied to the free running simulation do improve the prediction. Care is taken here as it is only a small out of sample test, and so we use this result in addition to those from the perfect model test.

L. 194: How are the temperature metrics influenced by the SD versus free-running simulations? What would happen if you calculated the individual and total weights (i.e.,Fig. 3) using Ref-C2 instead of Ref-C1SD?

Here we address the use of the refC1SD simulations. We did additionally construct weightings from both the refC1 simulations (essentially free running hindcasts) and refC2 during the conception of this study. As per our response to other reviewers, in our opinion the weights from these were not sensible for a number of reasons:
- The refC1 set of simulations is not large and not many models ran multiple realisations. As a result, we don't have a good cover over the possible variable space, meaning that only a couple of models received a strong performance weighting. The same issue was true for the refC2 simulations. This free running non-smoothed set up also meant that the dependence weightings are particularly useful because of the large range of model output, for example the range of the total column ozone values can be as large as 250DU which is almost 100% of the 'normal' ozone column.
- To remove the interannual variability, to allow for a fair comparison, we would have to smooth metrics to allow for a trend. Missing data makes the smoothing tricky and means that we lose an amount of data.

- Essentially, we wanted to give the models the best chance of replicating the observations and the way to do this was to use the specified dynamics runs. This allows us to use all the available observational data and to test model response to events which would be lost through smoothing, such as volcanic eruptions and sudden stratospheric warming events.

To justify the use of the weights generated using refC1SD on the refC2 simulations we performed both an out of sample test and a perfect model test (detailed in section 4.2). This shows that the weights learnt in one dynamical space are applicable to a different dynamical space.

And, similar to the previous comment, what if the nudging in the SD run is what causes a realistic decrease in temperature, not the coupling between decreased ozone and temperature (i.e., if ozone is poorly simulated, but the nudging imposes realistic temperature changes, will a high weighting be awarded to this model, for this metric, despite getting temperature "right for the wrong reasons")?

As for your comment about L148, we believe that the perfect model test goes some way to checking that the nudging is not overly influencing the weighting. For example, take a model which has strong temperature nudging but in a free running scenario performs poorly. We would generate weights highly favouring this model over others in the ensemble, but these weights would create a poor projection. In the perfect model test this behaviour would be apparent, because the weighted projection will compare badly to the pseudo truth (a model which we have taken as the truth in order to test the methodology). However, the improvement in the WM compared to the MMM (shown by the perfect model tests) suggest that the weights generated from the refC1SD dynamical system do not predominantly reflect the quality of nudging and can be applied.

L. 290: Are the results of this out-of-sample test shown anywhere? It's difficult for me to grasp what these RMSE values mean, in context, though I'd be curious to see the results of the test.

They have not, mainly because they don't make for a particularly informative graph, and it would just be a zoomed in repeat of Figure 2. That figure does show the WM and MMM, alongside the observations, although admittedly not the RMSE values. The quick plot below shows the trends of the observations, the weighted mean (WA) and the multi model mean (MMM) for the out of sample period, where the y axis is the total ozone column (DU) relative to 1980 values.

[Figure]

We have given this thought, but don't think this plot (or a similar one for RMSE) adds much to the manuscript, and so don't include it.

L. 336: On the sensitivity of the final weightings to which performance metrics are included: You performed a dropout test, leaving one metric out at a time. But, what about leaving out two? For instance, since CNRM-CM5-3 stands out so significantly in its Total Weight, what if you leave out the two metrics where that model apparently excels above the other models in performance, i.e., Polar vortex breakdown trend and Ozone-temperature gradient? I am concerned that the Total Weight for each model is highly sensitive to the combination of observational metrics included, beyond what is tested by the single-metric dropout test.

This is a good question and we understand your concern, considering the uneven distribution of the weights for some metrics. However, performing some additional analysis, we don't find the uncertainty in recovery date to be particularly sensitive to this. If you leave out additional metrics you get a very similar uncertainty in recovery date:
- 95% confidence interval of recovery for 1 model 1 metric dropout: [2052.4, 2060.4]
- 95% confidence interval of recovery for 1 model 2 metrics dropout: [2052.1, 2060.7]
- 95% confidence interval of recovery for 1 model 3 metrics dropout: [2051.8, 2061.1]

So up to a point, the recovery date uncertainty is quite stable. However, if we progress further than 3 metrics being dropped, we begin to lose the point of why we're doing a multi metric weighting in the first place.

From this we conclude that the recovery date is not particularly sensitive to the inputs and believe this shows a level of robustness. We have added this into the manuscript at L259.

L. 50: "...and climate" should be "and climate forcers" or similar?

Changed

L. 200: "high" should be "highly" and "except of" should be "except for"

Changed, thanks.

Yes, thanks!

 "The total weighting, formed from the summation of individual metric weights..."Is this right? This conveys to me that all of the individual metric weights are simply totalled. Since CNRM has a couple weights >0.4, then its resulting total weight of 0.27 suggests this is not right...Found by Eq. 2 instead?

It is mentioned in section 2 that we "normalised over all the models to sum to 1". For clarity we have changed summation to mean.

The total weighting, formed from the **mean** of individual metric weights **per model**, is largely influenced…

 "The lowest model weight is 55 % the value of a uniform weighting." I understand this to mean that the lowest red bar in Fig. 3 is 55% the magnitude of the dashed black line, but the smallest bars (CCSRNIES-MIROC3.2 and EMAC-L47MA) look to be less than half, judging by eye. Is this statement correct?

Good spot, thanks! This was a typo (should be 45%), which has been changed. We have also double-checked other numeric values.

[revised manuscript text omitted]

As noted in Knutti et al. (2017) there is not an objective way of determining optimal sigma values. Our method of selecting appropriate parameter values was to consider a training and a testing set of data, much like a machine learning problem. We

determined the values of sigma using the training data, which in this case is the refC1SD simulations, such that the weighted training data gave a good fit to the observations. The testing data (refC2 simulations) allowed us to test the weights and sigma values out of the temporal range of the training data, which avoids performing testing on data that was used to tune the parameters.

[revised manuscript text omitted]

370 from 0.5 hr to 50 hr and from varying reanalysis products (Orbe et al., 2020). We use the perfect model test to show that the utility of the weighting methodology is not compromised by using models with such a variety in nudging time-scales and methods.

As the perfect model test produces better projections, for models which are nudged in a variety of ways, we can conclude that

375 the weighting is not dominated by nudging. Take for example UMUKCA-UCAM which is nudged quite differently compared to the ensemble, as evidenced by a southern pole significantly colder than the ensemble. When we take UMUKCA-UCAM as the pseudo truth (temporarily assuming the UMUKCA-UCAM output is the observational truth) we generate weights based upon the refC1SD simulations and test them on the refC2 simulations. The weights generated are based on the dynamical system simulated in refC1SD which includes any model nudging. We can test how well these weights apply to a different

380 dynamical system without nudging (refC2). As we see an improvement in the WM compared to the MMM we can conclude that the weights generated from the refC1SD dynamical system can be applied to the refC2 dynamical system. If there hadn't been an improvement, then the dynamical systems described by refC1SD and refC2 may be too dissimilar for this weighting methodology and the weights may instead have been dominated by how well models are nudged. Nudging may be influencing the weights, but not to a degree that the accuracy of the projection suffers. Orbe et al. (2020) highlight the need for care when

385 using the nudged simulations and we would like any future work on model weighting to quantify the impact of nudging upon model weights to reflect this.

We justified using the nudged refC1SD simulations, despite these considerations, for two reasons. Firstly,  these nudged simulations give the models the best chance at matching the observational record, by providing relatively consistent meteorology across the models. The free running CCMI hindcast simulations (refC1) have a large ensemble variance and, despite

producing potentially realistic atmospheric states, are not directly comparable to observational records. Secondly, the perfect model testing discussed above, demonstrates that the nudging doesn't have a  detrimental effect on the ~~weighting. As the perfect model test produces better projections, for models which are nudged in a variety of ways, we can conclude that the weighting is not measuring nudging. Take for example UMUKCA-UCAM which is nudged quite differently, as evidenced by a colder pole than the ensemble. If the methodology was testing nudging, we would expect the perfect model test, when using UMUKCA-UCAM as the pseudo truth, to not produce a WM projection which was better than the MMM, because the nudging in UMUKCA-UCAM is not like any other model in the ensemble. However, this is not the case, and the WM projection is better than the MMM, confirming that the weighting is not largely dependent on the nudging process.~~model weighting.

Although we were not seeking to grade the CCMs as per Waugh and Eyring (2008), the construction of a weighted mean provides insight into model performance which would not be considered in a MMM. This is of some relevance as the CCMI ensemble has not undergone the same validation as its predecessors, such as CCMVal (Eyring et al., 2008). Additionally, we gain insight into model dependence shown in Sect. 4.2. Whilst this approach may not be as illuminating as Knutti et al. (2013), where they explored the genealogy of CMIP5 models through statistical methods, or Boé (2018), who analysed similarity through model components and version numbers, it successfully identified the known inter-model similarities. More complex methods are desirable, especially those that consider the history of the models' developments. Nevertheless, the simplicity of quantifying inter-model distances as a measure of dependence lends itself well to model weighting.

**6   Conclusions**

We have presented a model weighting methodology, which considers model dependence and model skill. We applied this over a suite of metrics grounded in scientific understanding to Antarctic ozone depletion and subsequent recovery. In particular we have shown that the weighted projection of the total ozone column trend, with inter-annual variability removed, predicts recovery by 2056 with a 95 % confidence interval of 2052–2060. Through perfect model testing we demonstrated that on average a weighted mean performs better than the current community standard of calculating a multi model mean. Additionally, the perfect model test, a necessary step in validating the methodology, showed a level of transitivity between the free running and the specified dynamics simulations.

[revised manuscript text omitted]

Orbe, C., Plummer, D. A., Waugh, D. W., Yang, H., Jöckel, P., Kinnison, D. E., Josse, B., Marecal, V., Deushi, M., Abraham, N. L., Archibald, A. T., Chipperfield, M. P., Dhomse, S., Feng, W., and Bekki, S.: Description and Evaluation of the specified-dynamics experiment in the Chemistry-Climate Model Initiative, 
[revised manuscript text omitted]